# Dual Role of Fibroblasts Educated by Tumour in Cancer Behavior and Therapeutic Perspectives

**DOI:** 10.3390/ijms232415576

**Published:** 2022-12-08

**Authors:** Belén Toledo, Manuel Picon-Ruiz, Juan Antonio Marchal, Macarena Perán

**Affiliations:** 1Department of Health Sciences, University of Jaén, E-23071 Jaén, Spain; 2Biopathology and Regenerative Medicine Institute (IBIMER), Centre for Biomedical Research (CIBM), University of Granada, E-18100 Granada, Spain; 3Instituto de Investigación Sanitaria ibs. GRANADA, Hospitales Universitarios de Granada-Universidad de Granada, E-18071 Granada, Spain; 4Department of Human Anatomy and Embryology, Faculty of Medicine, University of Granada, E-18016 Granada, Spain; 5Excellence Research Unit “Modeling Nature” (MNat), University of Granada, E-18016 Granada, Spain

**Keywords:** tumour microenvironment, cancer-associated fibroblast, cancer cell, magnification, inflammation, metastasis

## Abstract

Tumours are complex systems with dynamic interactions between tumour cells, non-tumour cells, and extracellular components that comprise the tumour microenvironment (TME). The majority of TME’s cells are cancer-associated fibroblasts (CAFs), which are crucial in extracellular matrix (ECM) construction, tumour metabolism, immunology, adaptive chemoresistance, and tumour cell motility. CAF subtypes have been identified based on the expression of protein markers. CAFs may act as promoters or suppressors in tumour cells depending on a variety of factors, including cancer stage. Indeed, CAFs have been shown to promote tumour growth, survival and spread, and secretome changes, but they can also slow tumourigenesis at an early stage through mechanisms that are still poorly understood. Stromal–cancer interactions are governed by a variety of soluble factors that determine the outcome of the tumourigenic process. Cancer cells release factors that enhance the ability of fibroblasts to secrete multiple tumour-promoting chemokines, acting on malignant cells to promote proliferation, migration, and invasion. This crosstalk between CAFs and tumour cells has given new prominence to the stromal cells, from being considered as mere physical support to becoming key players in the tumour process. Here, we focus on the concept of cancer as a non-healing wound and the relevance of chronic inflammation to tumour initiation. In addition, we review CAFs heterogeneous origins and markers together with the potential therapeutic implications of CAFs “re-education” and/or targeting tumour progression inhibition.

## 1. Introduction

According to the American Cancer Society, in 2022 there will be an estimated number of 1.9 million new cancer cases diagnosed and 609,360 cancer deaths in the United States. According to these data, cancer is, and will continue to be, one of the worst diseases in the world. Numerous people continue to pass away from cancer every year, despite major recent improvements in our understanding of the molecular and genetic causes of the disease [1]. This failure is primarily due to cancer cells’ resistance to current therapies, which causes tumour recurrence and metastasis, which account for more than 90% of cancer-related fatalities [2].

According to the “seed and soil” hypothesis, a suitable host microenvironment (the soil) is necessary for the best growth of tumour cells (the seed) [3]. Many researchers have concentrated primarily on tumour cells over the past 40 years. Emerging data suggest that tumours are made up of tumour parenchyma and stroma, two separate but interdependent components that intersect to support tumour growth. Cancer-associated fibroblasts (CAFs) are the most prevalent type of stromal cell in tumours [4,5] and are derived mainly from endogenous fibroblasts, apart from another great variety of cells, which are essential in maintaining homeostasis, controlling the growth and differentiation of neighbouring epithelial cells, and regulating wound healing processes in healthy tissues [6].

It is becoming more and more obvious that every tumour is different in some way, and that the way a tumour interacts with its surroundings has effects that appear to vary between tumour kinds and even between individuals with the same tumour type. There are still many unsolved issues about the connection between CAFs and the tumour, despite intriguing findings showing that CAFs depletion causes delayed tumour growth and enhanced response to treatment [7]. Do they originate as physiological barriers to fight against the aggression of the onset of the tumour mass? Are they allies of the tumour even before tumour cells begin to proliferate? Or are fibroblasts enlisted in the cancer crusade only after the call-up by tumour cells? Do normal fibroblasts react to cancer aggression as if it were a wound to heal within the tissue, and then become CAFs? Here, we will review the latest findings about the tumour microenvironment (TME) and carcinogenic events with the aim of discussing novel therapeutic approaches to help to fight cancer.

## 2. Cancer, a Non-Healing Wound

Dormant or resting fibroblasts are detected in areas of interstitial stroma between functional parenchymal layers in adult and well-differentiated tissues. They are described as thin, elongated cells with frontal and posterior extensions and fusiform shapes. From a molecular point of view, quiescent fibroblasts remain poorly defined [8,9]. In response to tissue injury, quiescent fibroblasts are activated into normal activated fibroblasts (NAFs), facilitating tissue repair and regeneration. NAFs were first described in wound healing and were predominantly identified by their expression of Alpha Smooth Muscle Actin (αSMA) and vimentin (VIM), along with enhanced collagens (type I, III, IV, and V), laminins, and fibronectin production, which remodel the extracellular matrix (ECM) [10,11]. Matrix metalloproteinases (MMPs) are also primarily produced by activated fibroblasts or myofibroblasts, highlighting their critical function in controlling ECM turnover and preserving ECM homeostasis [12]. In order to bind the borders of the wound, myofibroblasts constrict the ECM. They also produce matrix proteins, which fix any residual tissue flaws and draw epithelial cells to complete the healing process. Once this is finished, activated fibroblasts reduce in number and undergo a specific type of programmed cell death known as nemosis, which is fuelled by macrophages and fibroblasts that resemble macrophages [13,14]. As tumours are regarded as “non-healing wounds” [15,16] (due to the similarity of the process), in this regard, myofibroblast populations persist and initiate a chronic response known as cancerous or stromal fibrosis where tissue fibroblasts are fooled so that instead of undergoing apoptosis, the fibroblasts enhance proliferation and become a functionally diverse population known as fibrosis-associated fibroblasts (FAFs) or cancer-associated fibroblasts (CAFs) [17] (Figure 1).

Tumour stroma, in fact, shares many similarities with non-healing wounds, including inflammation, tissue formation, and remodelling [18]. In fact, inflammation is important in cancer progression because an inflammatory microenvironment can increase error and mutation rates while also promoting the proliferation of mutated cells [19]. Reactive oxygen species (ROS) and reactive nitrogen intermediates (RNI) (which are highly reactive chemicals formed from O_2_ and N_2_, respectively, and can easily react with other molecules in a cell) produced by activated inflammatory cells cause deoxyribonucleic acid (DNA) damage and genomic instability [20,21]. Furthermore, inflammatory cytokines such as Transforming Growth Factor β (TGF-β), among others, induce genomic instability and increase changes in tumour cell plasticity [22]. Inflammation also produces growth factors, cytokines like interleukins (IL-6), and a protein complex like NF-kB that can give tumour progenitors a stem cell-like phenotype or stimulate stem cell expansion [23,24]. Moreover, chronic inflammation has been proven to benefit the creation of a TME that attracts myeloid cells and supports tumour growth and angiogenesis [25,26,27]. In fact, epidemiological and clinical research has suggested that inflammation and tissue fibrosis in some organs, such as the liver, lung, and pancreas, may increase the risk of carcinoma development [28,29]. The conception that inflammation, or other forms of tissue aggression, can end up initiating tumour progression alters the generally accepted idea that tumour cells induce fibroblasts malignification and not vice versa (Table 1). In fact, an in vivo study demonstrated that injection of non-tumourigenic cells previously cocultured with CAFs led to the formation of large tumours and, in contrast, no tumour formation was observed when injected cells were co-cultured with NAFs [30,31].

## 3. Stromal Cells Are Recruited to the Battlefield by Cancer Cells

The cancer cell secretome, which includes factors such as TGF-β, platelet-derived growth factor (PDGF), and fibroblast growth factor 2 (FGF2), recruits stromal cells during tumourgenesis [48,49] (Table 1). The most abundant cells within the stroma are fibroblasts, which automatically activate the proliferative and carcinogenic Hedgehog and Notch pathways in response to tumour cell signalling [136,137] (Figure 2 and Figure 3). However, this mechanism is not universal, as the loss of Notch signalling can equally promote CAF phenotypes in squamous cell carcinoma [138]. Nonetheless, other mechanisms by which malignification of stromal cells is also achieved have been described, such as inflammatory modulators like IL-1 act through the NF-κB pathway and IL-6 activate transcription of STAT factors [8,46] (Table 1), the positive feedback process involving Janus kinase (JAK)-STAT signalling, the presence of a contractile cytoskeleton and alterations in histone acetylation [86,87], or the secretion of Galectin-1 during the epithelial–mesenchymal transition (EMT) process may also promote fibroblast transformation and induction of glioma cell migration [139] (Table 1). In addition, physical changes in the ECM [140], heat shock factor 1 (HSF1), which responds in part to protein misfolding [88,141], loss of suppressor genes such as PTEN, caveolin-1 (CAV-1), p53 and p21 [89,142,143], or physiological and genomic stresses such as double-stranded DNA breaks that induce the production of IL-6 and the TGF-β family ligand activin A can also favour this process [50]. It has been demonstrated that stromal fibroblasts exposed to carcinoma cells’ conditioned medium were more capable of promoting tumour growth [51,52]. According to Toullec and colleagues (2010), stromal cell-derived factor 1 (SDF-1) is a key factor involved in the malignancy and action of resident fibroblasts in human adenocarcinomas [144]. It is likely that carcinoma cells not only initiate the transformation of stromal fibroblasts into myofibroblasts but also contribute to their activation in vivo (Figure 2 and Figure 3).

Signals from other cells in the TME, as well as tumour cells, may instruct CAF function (Figure 3); for example, granulin produced by macrophages promotes the activation of a fibrotic environment in liver metastases. These mechanisms, which are not directly dependent on the presence of cancer, may contribute to the protumourigenic environments found in inflammatory conditions and are linked to an increased risk of cancer [145]. CAFs also cause tumour cells to secrete factors that stimulate them, such as hepatocyte growth factor (HGF), connective tissue growth factor (CTGF), epidermal growth factor (EGF), insulin-like growth factor (IGF), nerve growth factor (NGF), FGF, and Wnt family members [5,49]. In addition, CAFs secrete cytokines like CCL7, CXCL12, and CXCL14 that favour tumour progression [32,33,34,35]. An example of the “conversation” between tumour cells and CAFs comes from pancreatic ductal adenocarcinoma (PDAC), IL-6 derived from CAFs was modulated by retinoic acid, which was in turn associated with the EMT process of the tumour cells [36]. The positive activation of HSF1 in CAFs, with the consecutive enhanced production of TGF-β and SDF-1 in a non-autonomous manner is related to the malignancy of cancer cells [37,38] (Figure 3) (Table 1).

Furthermore, the fibrotic environment offers benefits for optimal tumour development, such as the formation of a physical barrier to immune surveillance, which allows for tumour growth [146]. CAFs also promote the development of a surrounding desmoplastic environment by synthesizing fibrillar collagens, which stiffens tissue and alters homeostasis, resulting in the collapsing of nearby blood vessels, which promotes hypoxia and thus results in more aggressive cancer phenotypes [147,148]. Furthermore, CAFs engage in paracrine communication with other cell types in the cancer ecosystem, inducing EMT in neoplastic cells [149] and increasing tumour invasiveness [150]. De Wever evaluated and demonstrated the proinvasive activity of human CAFs in vitro using human colon cancer cells and fibroblasts isolated from surgical fragments of colon cancer, further establishing the idea that cross-signalling between cancer cells and fibroblast-like stromal cells may contribute to the progression of human solid tumours [151].

Although all fibroblasts present in a tumour are defined as CAFs, they come from diverse cell sources [115]. We have already described above how resident fibroblasts are activated to become CAFs, but other cell types within the tissue also undergo a transformation that ends in an induced cell differentiation toward CAF-like cells. It is becoming evident that the origin of CAFs may vary between different tumour histotypes, and within different areas of the same tumour, and this origin may be critical in determining the degree of CAFs malignancy [5,152,153] (Figure 4).

Many studies support the idea that mesenchymal stem cells (MSCs) are recruited and induced by the developing tumour to enhance proliferation and to acquire a CAFs-like phenotype [154]. MSCs malignification is driven by cytokines and chemokines released in the TME such as TGF-β, PDGF, and IL-6, among others [4,39] (Table 1). For instance, bone marrow (BM)-derived human MSCs exposed to tumour-conditioned medium differentiate and express myofibroblast markers αSMA and fibroblast activation protein-alpha (FAPα) [155]. Based on studies in animal models, such as mouse gastric cancer, it has been estimated that BM-derived MSCs can account for up to 15–25% of the CAFs population [156]. In another study, after 28 days of implantation of pancreatic carcinoma cells in a murine tumour xenograft model, BM cells accounted for nearly 40% of the total stromal myofibroblast population within the tumour [157]. Intriguing studies in patients who had previously received a BM transplant from an opposite sex donor revealed that a high percentage of the tissue fibroblasts were of donor origin [158]. In contrast, Arina et al. (2016) recently showed that CAFs were derived primarily from local fibroblasts, not from cells in the BM, thus supporting the theory of the multiple origin of CAFs (Figure 4).

A recent study found that loss of E-cadherin in tumour buds increased VIM expression and CAFs activation, all of which are signs of EMT and are associated with more aggressive tumours [159]. CAFs, on the other hand, can develop from endothelial cells via the endothelial-to-mesenchymal transition (EndMT). They undergo partial transdifferentiation, acquiring fibroblastic markers like αSMA, FSP-1, VIM, and N-cadherin while retaining endothelial markers like VE-Cadherin, CD31, and Tie1/2. Furthermore, the absence or reduction of adipocytes in pathological tissue could be due to activated fibroblasts interfering with their differentiation process. Adipocytes may also play a role by interacting with cancer cells and providing metabolic support, whether or not they are converted to CAFs [160] (Figure 4).

## 4. CAFs Biomarkers: A Mixed Bag

We can agree that a clear correlation exists between tumour initiation, fibroblast corruption, and tumour progression, so identifying CAFs markers could help to determine the stage of tumour development and the initiation of the pre-metastatic niche (Figure 5). However, a vast variety of proteins associated with CAFs that present issues when it comes to being biomarkers with potential clinical applicability have been described. In fact, CAFs markers are expressed at different intensities at different cancer stages and this great heterogeneity has led to the realization that there are different CAFs subpopulations with key roles in the processes of tumour development and spread [161,162]. Here, we review CAFs markers that are commonly accepted and their potential therapeutic implications.

### 4.1. Cytoskeleton and Cytoplasmic Proteins

In the first place, we identify αSMA, a cytoskeleton protein associated with TGF-β production, with a highly contractile phenotype [163], which is considered a robust marker of CAFs possessing myofibroblast morphology [164]. Nevertheless, although αSMA is a marker commonly used to determine the presence of CAFs, the fact that it is also present in normal fibroblasts, even with expression levels comparable with those of CAFs [165,166,167], and in pericytes or myocardiocytes [168], implies a certain degree of confusion (Figure 4). To introduce more complexity, the degree of expression of αSMA seems to be correlated with different CAFs subtypes. Tuveson and colleagues distinguished two groups of CAFs in co-cultures of murine pancreatic stellate cells (PSCs) and human PDAC organoids. CAFs that were located next to tumour cells showed a contractile matrix-producing phenotype and a high expression of αSMA (myofibroblast-like CAFs or myCAFs). On the other hand, they noted that when CAFs were found in regions far from the tumour focus, the cells presented an inflammatory-like phenotype (iCAFs). Those iCAFs showed an immunomodulatory secretome with lower expression of αSMA and a high IL-6 secretory capacity through which they activate JAK/STAT [62]. Interestingly, CAFs subpopulations could transform into each other depending on the spatial or biochemical niche where they were cultured. For example, iCAFs grown in monolayers reverted to a myCAFs phenotype and expressed myCAFs markers [169]. CAFs plasticity implies powerful therapeutic implications, as it could mean that an active fibroblast may revert to an inactive form (Figure 6). JAK inhibitors can suppress tumour growth as well as the switch between iCAFs and myCAFs, whereas TGF-βR inhibition could partially attenuate myCAFs function without influencing tumour growth [40,41,42]. Nevertheless, both subtypes seem to influence tumour progression and hamper common therapies. While iCAFs appear to promote tumours by producing chemokines and cytokines [43], being markers of malignancy in pancreatic tumourigenesis [41], myCAFs increase ECM deposits impeding drug delivery [40].

Another highly accepted marker is the fibroblast-specific protein 1 (FSP-1 or S100A4). This marker is also not only limited to CAFs but can be found in epithelial cells undergoing EMT [170,171] and BM-derived cells, such as liver macrophages, that respond to tissue injury [172]. Subpopulations of S100A4+ fibroblasts present in TME have been shown to facilitate malignant progression, for instance a study with FSP-1-deficient mice showed reduced tumour growth and attenuation of metastatic potential, whereas injection of FSP-1+ CAFs partially reversed this effect [173]. S100A4+ CAFs promote tumour metastasis through secretion of VEGF-A and Tenascin-C (TN-C) [90].

Further, Vimentin (VIM), a type III intermediate filament protein that maintains cell integrity and is involved in cell migration, motility, and adhesion [174,175], which has been described as a CAFs marker. In fact, it has been shown that VIM plays a functional role in lung adenocarcinoma invasion and metastasis [176]. However, VIM’s efficacy as a CAF-specific marker is hampered by its widespread expression in both the general fibroblast population and in a variety of other cell types, including macrophages, neutrophils, lymphocytes, and any other cells of mesenchymal origin [177].

### 4.2. ECM-Related Components

Tenascin-C (TN-C), a member of ECM glycoproteins, is markedly up-regulated in pathologic tissues that are undergoing remodelling, such as those that are suffering inflammation, wound healing, or tumour progression [135,178]. In fact, negative prognoses of breast, colon, liver, and oral squamous cell carcinoma lymph node metastases cancer patients have been correlated with TN-C expression [179,180,181]. New results suggest that TN-C promotes multiple events in tumour progression and metastasis and, in particular, in angiogenesis [182]. Thus, specifically targeting TN-C could be a possible way to prevent the establishment of pre-metastatic niches [183].

Periostin (POSTN), a matricellular N-glycoprotein, is implicated in the EMT process, and is crucial for the collagen fibrillogenesis process [184,185] and wound healing [186,187]. Cellular survival, motility, and adhesion are facilitated by POSTN-activated signalling pathways, which are essential for tumour growth, angiogenesis, invasion, and metastasis [188,189,190]. Studies have showed that POSTN is predominantly expressed in tumour stroma [191,192,193]. Therefore, this protein could be a suitable CAFs marker.

The tissue inhibitor of metalloproteinase-1 (TIMP-1) has been shown to co-express in fibroblasts with α-SMA, and thus is a candidate to become a CAFs marker [100]. In fact, TIMP-1 released by CAFs promotes the migration and spread of cancer cells [101]. In lung adenocarcinoma, it was demonstrated that SMAD3/TIMP-1 in CAFs and CD63 in cancer cells were necessary to drive tumour progression in vitro and in vivo [102]. In another study, CAF-derived TIMP-1 promoted the migration of a colon cancer cell line, whereas TIMP-1 neutralization inhibited the enhanced migration, and TIMP-1 secretion was higher in CAFs co-cultured with cancer cells than in monocultured CAFs [103] (Table 1).

Osteopontin (OPN), a chemokine-like phosphorylated glycoprotein released intracellularly, is typically elevated in a variety of human malignancies [104]. It is essential for cell migration, EMT, ECM invasion, and in maintaining the stemness of tumour cells by either binding to integrins and CD44 or by activating the NF-Kb, MEK/MAPK, PI3K/Akt, and FAK pathways [44,79,105,106] (Figure 2 and Figure 3). Additionally, it has been shown that OPN plays a significant role in the differentiation of myofibroblasts into cardiac and dermal fibroblasts in culture, which may be pertinent to the fibrotic process [107,108]. Breast cancer cell-derived OPN promotes the activation of resident fibroblasts into CAFs by the association with CD44 and αvβ3 integrin on the fibroblast cell surface, which mediates signalling through Akt and ERK to induce Twist1-dependent gene expression [109,110] (Figure 2 and Figure 3). Furthermore, tumour-derived OPN also transforms MSCs into CAFs through the transcription factor, myeloid zinc finger 1 (MZF-1)-dependent TGF-β1 production, promoting tumour growth and metastasis [53] (Table 1).

### 4.3. Receptors and Membrane-Bound Proteins

The next candidate is FAPα, a type II integral membrane serine protease of the prolyloligopeptidase family. FAPα has been reported to be overexpressed in CAFs in many types of carcinomas, including colorectal, ovarian, breast, bladder, and lung carcinomas [111,194]. FAPα overexpression has been associated with poor prognosis as occurs, among others, in rectal [195], hepatocellular [196], colon [197], and pancreatic cancers [198]. Although FAPα is not solely specific to CAFs, as it has also been observed in bone, BM stem cells, and skeletal muscle [199,200], it could be one of the most promising therapeutic targets (Figure 6). FAPα silencing has been shown to inhibit stromagenesis, tumour growth, and angiogenesis in murine models of pancreatic cancer [201] and to suppress cell proliferation, migration, and invasion of ovarian squamous cell carcinoma cells in vitro and in vivo after inactivation of phosphatase and tensin homolog (PTEN), phosphoinositide-3-kinase (PI3K), and serinethreonine kinase. The use of immune checkpoint antagonists that promote T-cell function, such as anti-cytotoxic T lymphocyte-associated protein 4 (α-CTLA-4), and anti-programmed cell death 1 (α-PD-L1), has revealed an unexpected benefit in PDAC therapy based on FAPα+ CAFs depletion [199] (Figure 6).

Platelet-derived growth factor receptors (PDGFR) are cell surface tyrosine kinase receptors for members of the PDGF family. PDGFRα and PDGFRβ are high molecular weight membrane proteins (around 180 kD) which dimerize non-covalently upon ligand binding. PDGFRα is a marker for fibroblasts in some tissues like skin [65], whereas PDGFRβ is expressed in fibroblasts, pericytes, and CAFS in multiple tissues and tumour types [66]. A blockage of PDGFR signalling has been shown to suppress angiogenesis and tumour growth and a case in point is Dasatinib, a PDGFR inhibitor, which can partially reverse the pro-tumorigenic effect of CAFs in lung adenocarcinoma and may be a potential treatment strategy [67]. Based on their ability to promote the growth of co-injected pancreatic tumour cells in immunocompromised mice, PDGFR+ CAFs have protumourigenic properties in vivo [68] (Table 1).

### 4.4. Growth Factors and Cytokines

SDF-1 is a homeostatic chemokine that is also known as CXCL12 (Figure 2 and Figure 3). Fibroblasts in invasive breast carcinomas participate in tumour development primarily by secreting SDF-1 [92]. High levels of SDF-1 in endothelial cells and fibroblasts within ischemic injury have been shown to attract circulating CXCR4-expressing progenitor cells to injury sites [93,94]. Thus, elevated expression of CXCR4 by carcinoma cells favours the primary tumour in murine xenograft models [95], whereas deletion of this marker in breast carcinoma cells inhibits tumour growth [96]. Further, over-expression of CXCL14 and CXCL12 chemokines by myofibroblasts enhances epithelial cell proliferation and invasion, demonstrating the important paracrine effect of stromal cells on carcinoma cells [96,97].

Finally, Prostaglandin E2 (PGE2), known to be produced by immune cells, is also produced by malignant stromal cells like CAFs, and it promotes angiogenesis, induces metastasis, and increases tumour cell survival and proliferation [202,203,204]. It has been shown that PGE2 stimulated VEGF-A production in stromal fibroblasts generated from mice’s stomach cancer [51]. In addition, PGEP2 and PGEP4 are expressed in lung fibroblasts [205,206] and in CAFs from CRC to stimulate the development and expression of genes associated with inflammation and carcinogenesis [207].

## 5. Fibroblasts: Induced and Inductors

Normal fibroblasts appear to have a double effect on cancer cells. Fibroblasts act against malignant progression as early as during tumourigenesis [208,209], but as cancer progresses, fibroblasts are subverted to promote tumour growth (so-called CAFs) [97]. In fact, during the early phase of tumorigenesis, non-CAF fibroblasts have a protective role against tumour progression, enhancing intercellular induction of tumour cell apoptosis through four different pathways mediated by ROS (NO/peroxynitrite pathway, HOCl pathway, NO consumption, and nitrogen chloride pathway) [210]. However, if tumour cells scape this fibroblast “first line of defence” mediated by ROS, they proceed to malignify stromal elements and to favour a cross-communication between fibroblasts and tumour cells, which leads to tumour progression. Although the reasons for CAFs remaining activated indefinitely are unknown, it is clear that fibroblasts engage in reciprocal and elaborated feedback with cancerous epithelium and the dual action of fibroblasts during tumour development [211].

Interestingly, a recent study found that CAFs are capable of reverting lung cancer spheroids from a dysplastic to a hyperplastic state by knocking out Sox2 activity. Therefore, CAFs are capable of modifying intrinsic oncogenic characteristics and influencing tumour development [212]. SLITs are a family of secreted ECM proteins that act as tumour suppressors under normal conditions and play an important role in cell migration, tissue development, and vascular network establishment. At first, when a fibroblast is in the early stages of activation, SLIT2 increases and inhibits the expression of CXCL12, leading to the prevention of tumour growth and metastasis. However, CAFs decrease SLIT2 production, and subsequently, up-regulate CXCL12, promoting metastatic behaviour of tumour cells, which supports the theory of aforementioned functional duality in fibroblasts [98,99].

Injecting a teratoma cell line into an early mouse embryo demonstrated that healthy cells have tumour suppressor properties; the resulting postnatal animals were derived from both host cells and transplanted tumour cells, but the mouse presented normal development with no sign of specific features of tumour cells [213,214]. Dormant fibroblasts were assimilated to tumour suppressor cells because they inhibited the growth of in vitro-transformed baby hamster kidney cells and in vivo-transformed keratinocytes [215]. Tumour growth suppressor factors, such as the WAP Four-Disulfide Core Domain 1 (WFDC1) protein, were found to be highly expressed in resting fibroblasts but downregulated in CAFs [216], implying that resting fibroblasts can indeed inhibit tumourigenesis. It is unclear whether these fibroblast subtypes are normal fibroblasts resistant to CAF conversion or distinct subpopulations of antitumour CAFs. Tumour suppressor fibroblasts have also been found in established tumours, according to research [217]. PDAC is known for having a large amount of fibrotic tissue within the tumour mass, as the majority of the tumour volume is made up of αSMA+ fibroblasts [217]. The removal of these fibroblasts was thought to reduce the aggressiveness of PDAC, but clinical studies using this strategy revealed accelerated disease progression [147]. Further investigation in mice revealed that deletion of αSMA+ fibroblasts resulted in invasive tumours with increased EMT and CSCs sub-population, resulting in a decrease in survival. This result was mainly due to decreased immune surveillance and increased regulatory T (Treg) cells infiltration. Thus, it seems that the PDAC-associated fibrotic response is a primary host defence mechanism against tumour invasion which then undergoes a malignification process [38].

In solid tumours, not only do CAFs remain persistent and autocrine activated, but they also stimulate neoplastic cells in a paracrine manner [218,219], accompanying tumour cells throughout the carcinogenic process [112,220] (Figure 5). CAFs regulate epithelial proliferation as a result of their direct paracrine interactions with tumour cells, having an indirect effect on inflammatory processes in which CAFs serve as mediators. Interestingly, induction of CAFs autophagy has been shown to promote tumour cell survival through processes involving the downregulation of CAV-1 and subsequent stabilization of Hypoxia Inducible Factor 1-alpha (HIF1-α) [221].

CAFs increase the secretion of MMPs as observed in studies of head and neck squamous cell carcinoma (HNSCC), highlighting their crucial role in cancer progression and early stages of tumour growth [49]. For example, MMP-13 promotes tumour angiogenesis and leads to tumour invasiveness [222]. TME stromal collagen accumulation is associated with poor prognosis in primary tumours and metastases [223,224,225]. Desmoplasia reduces microvascularity, limiting chemotherapy treatment penetration [226,227,228]. In fact, in a mouse model of pancreatic adenocarcinoma, lowering CAFs improved chemotherapy efficacy, implying that CAFs play an important role in intratumoural drug delivery in function of its ECM secretion [229] (Figure 6). Attempts to target the stroma, either directly or by enzymatically digesting ECM, have resulted in reduced tumour growth and improved chemotherapy response [228].

The role of tumour stromal fibroblasts in carcinogenesis was studied using an immunodeficient mouse tumour xenograft model in which tumour cells were implanted alongside CAFs, homologous fibroblasts extracted from non-cancerous tissue from the same individual, or normal fibroblasts extracted from healthy donors [230]. Tumours formed from carcinoma cells injected with CAFs grow faster than tumours formed from any of the other fibroblast types.

In addition, CAFs support invasion and metastasis by modelling the ECM [231], increasing tumour spread and therapeutic resistance [232]. Indeed, in primary tumours, IL-11 secretion by CAFs favours neoplastic GP130/STAT3 signalling, reprogramming MSCs into CCL5-secreting CAFs, and finally leading to cell invasion and promotion of metastasis [47] (Figure 5). In addition, tumour progression is enhanced by CAFs IL-8 secretion [233], CAFs exosomes autocrine induction of Wnt-11 [234], potentiation of the neoplastic EMT process for tumour extravasation [149], or by CAFs MMPs secretion to trigger the COX2-NF-B cascade, as seen in prostate cancer cells and promotion of ROS accumulation and invasion [235] (Table 1).

Interestingly, the key role of CAFs as chaperons in metastasis is becoming more widely accepted. CAFs facilitate intravasation and extravasation and escort and sustain neoplastic cells within the blood stream on their way to metastatic sites [173,236,237] (Figure 5). Recent in vivo studies have shown that metastatic cells have greater viability when they are accompanied by their own stromal elements, while when CAFs are depleted, there is a marked decrease in the number of lung metastases [238]. Invasive tumours in colon cancer patients have also been shown to be enveloped by a layer of CAFs during invasion [239]. Aiello et al. demonstrated that fibroblasts appear in distant metastases with 6-7 cells and that the goal of the final metastasis is to reach a stromal volume of the dimensions of the primary tumour [240].

Furthermore, major mediators of breast cancer metastasis to the lung include metastasis-associated fibroblasts (MAF) expressing FSP-1, TN-C, and VEGF-A [91]. The activation of liver-resident fibroblasts to enhance angiogenesis is necessary for melanoma spread to the liver [241]. Loss of this protein in FSP1 mutant mice affects fibroblast motility and is linked to decreased metastasis [242]. For CAFs extracted from prostate carcinoma samples, a similar pro-metastatic impact was also shown [82], where CAFs induce lung metastasis after heterotopic co-injection of cancer and stromal cells. Additionally, CAFs can boost the rate of successful colonization by choosing neoplastic clones that have a particular advantage over metastatic spread while still present in the initial lesion. CXCL12 and IGF secretion serve as mediators in this process [83] (Figure 5) (Table 1).

Finally, CAFs’ support to CSCs has also been proven. Chen et al. recently found that CAFs, through IGF-II and IGF1-R intercommunication, can improve lung cancer root plasticity through mechanisms dependent on Nanog [84], a transcription factor involved in CSC pluripotency. Giannoni et al. reported that CAFs isolated from prostate carcinoma samples, through the activation of the EMT epigenetic program, promote CSC generation [82], and increase the expression of recognized CSC markers (CD133+ and high CD44/CD24 ratio) and the creation of nonadherent spheres as an associated property [85]. Others were able to demonstrate increased cell proliferation and elevated EGFR expression in HNSCC tumour 3D spheroids when co-cultured with CAFs, in addition to an EMT and CSC-like phenotype, and decreased sensitivity to the cytotoxic agents, cisplatin and cetuximab (Mustafa Magan et al., (2020)) (Table 1).

## 6. CAFs and Tumour Immunity

CAFs are beginning to take the lead rather than play a supporting role in immune modulation in recent years. The CAFs secretome may vary dynamically during the evolution of cancer by supporting tumour immunity at various stages of the tumour, but it does not contribute to a state of prolonged activation of the defensive level [243,244].

CAFs can affect the host immune response both directly and indirectly, according to Knops et al.’s (2020) study. By sustaining an abnormal pro-tumour inflammatory milieu, they can hinder or abolish the cytotoxic actions of T and NK cells [245,246]. In order to help create and maintain an immunosuppressive milieu that impairs adaptive immunity against the tumour, CAFs promote the recruitment, development, and survival of regulatory T cells [247,248]. Cytotoxic T lymphocytes that are specific for an antigen are eliminated as a result of the presentation of Human Leukocyte Antigen (HLA)-class I antigens by stromal fibroblasts and the production of PD-L2 and FASL [249]. Cytotoxic T cell activity was restored, and fast tumour necrosis took place in genetically engineered Lewis lung cancer (LCC) mice when FAPα+ stroma was specifically removed by diphtheria toxin [250]. The doxorubicin-based chemotherapy reduced the invasion of immune-suppressive cells like regulatory T cells (Treg) and further stimulated the anti-tumour immune response when this deletion was used with it [251].

The subtype known as CAFs-S1, which is identified by the expression of FAPα, αSMA, PDGFR, and CD29 markers, was found to be associated with the recruitment, retention, and differentiation of Treg cells, enhancing their capacity to inhibit CD4+CD25+ T-cell proliferation in vitro. This was revealed by a FACS-based analysis of CAFs in human breast tumours. Additionally, it was able to prolong T-cell life and trigger their differentiation into regulatory lymphocytes with the CD4+CD25+FOXP3+ phenotype [69]. This process was also revealed to be dependent on CXCL12’s differential regulation by miR-200/141 in human high-grade serous ovarian tumours (HGSOC) [70].

More recently, it has been observed that CAFs can lead to activation of CD4+ T cells and suppression of CD8+ T cells [252]. Targeting FAPα+ CAFs demonstrated antitumour effects through intratumour recruitment of T lymphocytes in genetically engineered animal models and orthotopic cancer transplant trials in immunocompetent mice [63,253]. The absence of FAPα+ CAFs cells appears to restore immune detection and tumour destruction [251] (Figure 6).

CAFs-derived CXCL14 affects macrophage recruitment to the tumour [254] and shunting of their differentiation status toward the M2 subtype is associated with a bad prognosis in cancer. In a mouse model of lung adenocarcinoma, Hegab et al. also noted that CAFs promoted the conversion of tumour-associated macrophages (TAMs) to the M2 phenotype [255]. Additionally, it has been demonstrated that the production of the chitinase-like protein 3 (Chi3L1) by mammary CAFs induces the recruitment of macrophages with an M2-like phenotype and reduced CD8+ T lymphocyte infiltration [256]. Additionally, in numerous animal cancer models, including colorectal, prostate, and breast carcinomas, CAFs have been demonstrated to recruit macrophages in TME [257]. Additionally, tumour-educated CAFs (derived from MSCs) attracted CD11b+Ly6C+ monocytes, F4/80+ macrophages, and CD11b+Ly6G+ neutrophils via the CCL2-CCR2 axis to promote tumour immunity in a mouse model of spontaneous lymphoma [113].

In a mouse model of transplantable colorectal carcinoma (CRC), it was found that fibroblasts expressing high levels of FAPα recruited myeloid cells via CCL2, which resulted in resistance to anti-PD-1 immune checkpoint therapy that was eliminated by targeting the FAPα. This is significant because it indicates that myeloid cell recruitment by CAFs is also associated with resistance to therapy. These results were confirmed in tissue samples from human CRCs, where the presence of fibroblasts with high FAPα+ concentration was positively connected with myeloid cell infiltration and negatively correlated with T-cell infiltration [258,259]. Similar effects were shown when FAPα was pharmacologically targeted in a transplanted pancreatic cancer model, where there was a decrease in macrophage recruitment and an increase in T-cell infiltration [260]. Additionally, in transgenic and transplanted mice models of melanoma, targeting FAPα+ fibroblasts by immunization with an adenoviral vector eliminated the recruitment and function of immunosuppressive cells, including monocytic and polymorphonuclear MDSCs, within the TME [261]. Human prostate tumour CAFs were discovered to secrete SDF1 to draw monocytes into primary cultures in vitro. Additionally, the production of the immunosuppressive cytokine IL-10 increased due to the M2-like polarization of circulating monocytes that was induced by these SDF1-producing CAFs [45]. These results are in line with the functional role that CAFs-derived SDF1 has been shown to play in supporting tumour development and immunosuppression. In addition to secreting metabolic reprogramming molecules such as indoleamine 2,3-dioxygenase 1 (IDO1), arginase 2 (Arg2), and galectin, the αSMA+ CAFs (myCAFs) also induce T-cell anergy and limit CD8+ T-cell proliferation [262,263,264].

## 7. Tumour Stroma and Therapeutic Resistance

Tumour stroma is not only implicated in all stages of cancer development and its immune resistance, but it can also function as an interstitial barrier, preventing effective drug delivery and promoting radioresistance [265,266]. Fibroblasts can be of influence by either (i) sustaining specific subtypes of neoplastic cells, such as CSCs [267], (ii) rewiring the signalling networks in cancer cells [80], or (iii) modulating immune cells [268]. All this leads us to the conclusion that both the presence and the amount of CAFs will be determinants in the final outcome in terms of tumour progression, dissemination, and chemoresistance [269].

CAFs’ immunological regulation, proangiogenic effects, and metabolic reprogramming might promote cancer cell survival and make it easier for the disease to evade the effects of therapy. For the first time, it was discovered in 2009 that breast cancer patients with high stromal gene expression did not respond well to neoadjuvant chemotherapy because of an elevated intratumour interstitial fluid pressure that subtly reduced antineoplastic medication uptake [270]. Notably, adhesion of tumour cells to CAFs, through homotypic binding of N-cadherin, can induce EMT progression, leading to therapeutic resistance, among other methods [271,272]. In turn, cancer cells promote fibroblast proliferation through oxidative stress and fibroblasts protect tumour cells from apoptosis through autophagic mechanisms [273]. Worryingly, standard chemotherapy has been shown to transform mammary NAFs into CAFs, resulting in a hypoxic, glycolytic, autophagic, and proinflammatory TME that in turn stimulates Sonic hedgehog (SHH)/GLI signalling, an antioxidant response, and interferon-mediated signalling in breast cancer cells. [114].

Additionally, CAFs’ released soluble components have been linked to resistance to anticancer treatment therapy. One example is the HGF, which has been linked to prostate cancer and is thought to play a critical role in CAF-mediated resistance to therapy with tyrosine kinase inhibitors or EGFR receptors [80,81] (Figure 6). Colorectal cancer resistance is mediated by this factor, which then activates the HGF receptor c-Met [274]. According to Crawford et al. (2009), CAFs play a critical role in the resistance to anti-angiogenesis therapy because PDGF expression is also positively regulated in tumour CAFs that are resistant to anti-VEGF therapy and administration of a PDGF-neutralizing antibody was able to change this anti-VEGF resistance (Figure 6). Additionally, TGF-β1 promotes resistance to docetaxel in prostate cancer through induction of the anti-apoptotic gene Bcl-2 by acetylation of KLF5 and to oxaliplatin in human colorectal cancer through EMT [54,55,56]. It can also reduce a tumour’s response to α-PD-L1 blockade by limiting CD8+ T cell accumulation in TME [57]. In colorectal cancer, TGF-β2 and HIF1 produced by CAFs can also activate GLI2, a mediator of the Hedgehog signalling pathway, to encourage the growth of cancer cells and chemotherapy resistance [58]. Finally, non-small cell lung carcinoma (NSCLC) is less sensitive to cisplatin when IL-6 produced by CAFs stimulates well-studied pro-survival signalling cascades and enhances TGF-β1-driven EMT in lung cancer cells [59].

## 8. Therapeutic Strategies

The idea that there are several factors involved in chemo/radiotherapy resistance that go beyond the tumour cells is on the rise. The physical barrier implied by the TME, with not only the proteins that form the stroma but with the interactions of the different cells that play a role on it, is a key example in which CAFs are protagonists. Thus, a detailed and personalized characterization of CAFs and stromal profiles is essential for the detection of novel targets to optimize therapeutic strategies while maintaining the survival of healthy cells and preventing the emergence of resistance [7]. Since the stromal and epithelial components of a tissue, malignant or healthy, act in an integrated and reciprocal manner, combination therapy might be an effective treatment choice. In conclusion, general CAFs depletion is probably not the best strategy for anti-CAFs therapies, while the reversal of the CAFs phenotype to a “non-CAFs” phenotype or inhibition of CAFs protumourigenic signals might instead offer better therapeutic options [275,276] (Figure 6).

Recently, new therapeutic strategies aimed at “re-educating” CAFs into a non-tumour associated status have been developed (Figure 6). The use of epigenetic modulating agents such as the demethylating agent 5-Azacytidine (5-AZA) has been shown to transform an aggressive PDAC model into a less aggressive and drug-sensitive phenotype, slowing tumour growth in vivo, after the engraftment of treated transformed cells. Gemcitabine (GEM) was then added to 5-AZA-treated cells, and this had a noticeable growth inhibition effect on GEM-resistant pancreatic cancer cells [277] (Figure 6).

Other approaches consist of the use of TGF-β inhibitors. The dual role of TGF-β in tumour progression is clear, acting as a suppressor at early stages and as a promotor later on. This functional duality could be demonstrated by comparing the effect of TGF-β signalling pathways in tumour cells of transgenic mice where it suppressed initial tumourigenesis but promoted the final metastatic process [38]. Detailed above is how the presence of TGF-β in TME promotes tumour growth by enhancing stromal support through the stimulation of CAFs collagen production (also preventing T-cell access to the tumour site), angiogenesis, and altering immune surveillance [60]. Thus, the use of molecules that suppress TGF-β action is a therapeutic option that is being studied. Currently, antibodies against TGF-β, TGF-β inhibitors (such as LAP or TGF-β kinase inhibitors), or TGF-β antisense oligonucleotides have been developed, many of which are in phase II trials. Preliminary results show a reduction of blood vessels and blood lacunae in tumours, and a decrease in microvessel density together with wet weight of treated tumours [61].

Recent research has examined the use of miRNAs as a potential method to deactivate CAFs [116,117] (Figure 6). MiR-19a-19b-20a was shown to be up-regulated in pulmonary fibrosis as the condition worsened. This miR inhibited TGF-β-induced fibroblast activation, which may indicate that healthy fibroblasts try to prevent themselves from becoming activated to a malignant phenotype [118].

On the other hand, miR-145, an inhibitor of the TGF-β pathway, was upregulated in both CAFs produced by TGF-β signalling and CAFs from primary oral cancer, but not in healthy fibroblasts. Prior to TGF-β therapy, normal oral fibroblasts were transfected with miR-145, which inhibited their activation to the CAFs phenotype. After being converted to CAFs by TGF-β, they were transfected with miR-145, and this resulted in the restoration of their quiescent phenotype, indicating that normal fibroblasts may already have a built-in negative feedback loop to guard against CAF conversion [119]. Another study found that when compared to fibroblasts from healthy breast tissue, CAFs from human breast tumours showed lower levels of the tumour suppressor miRNA Let-7b. It was shown, in a murine breast cancer model, that Let-7b miRNA inhibition, in normal mammary fibroblasts, increased their malignification, which in turn potentiated EMT induction and enhanced tumour growth. The opposite was observed when Let-7b was over-expressed in human breast CAFs, which ended in a reduction in cancer promotion [278].

Another therapeutic approach has been focusing on therapies to inhibit paracrine signalling of CAFs, for example, inhibitors of the HGF/MET signalling pathway, hedgehog, or angiotensin II receptor showed a reduction in the expression of CAFs and ECM content, thus enhancing drug delivery and inhibiting tumour growth and metastasis [279] (Figure 6). HGF is secreted by fibroblasts to activate c-Met in cancer cells [280]. MET kinase inhibitors have been shown to produce a beneficial effect in NSCLC [281,282]. Other preclinical studies focused on the use of NK4, a competitive Met antagonist [279] (Figure 6). Phase I and II clinical trials using innovative small compounds that target MET tyrosine kinase have been carried out in this regard, including tivantinib combined with GEM [283], cabozantinib [284], and crizotinib [285], concluding that HGF/MET-targeted therapy could be a promising therapeutic option [286].

The stromal tumour feedback requires the overexpression of PDGF, as was already mentioned. Additionally, as shown by gene targeting experiments, this signalling also supports angiogenesis in an autocrine and paracrine manner [40,71], and its expression is correlated with increased vascularity and vascular wall maturation [72]. It has been demonstrated in experimental models that PDGF inhibition decreases interstitial fluid pressure in tumours and improves the impact of chemotherapy [73,74]. As a result, PDGFR-specific antibodies may be utilized to prevent angiogenesis in tumours that do not respond to anti-VEGF therapy [75] (Figure 6). A powerful inhibitor of the VEGFR, PDGFR, and FGFR families, the indolinone derivative BIBF1120 is currently being used in clinical settings [76]. For instance, blocking PDGF receptor activation in CAFs in a mouse model of cervical carcinogenesis prevents the formation of premalignant lesions and chemoresistance [77,78].

Inhibiting the mTOR pathway in pancreatic CAFs with the somatostatin analogue SOM230 results in reduced secretion of IL-6, which stimulates chemoresistance, mediated by CAFs, and potentiated pancreatic tumour drug sensitivity [64]. Other therapeutic targets have also demonstrated the reduction of chemoresistance (Figure 6). Targeting tumour stromal elements necessary for tumour progression, such as fibronectin, can also successfully slow down tumour development and spread while improving antitumour medication delivery. In fact, according to recent research, fibronectin depletion causes the secreted protein acidic and cysteine rich (SPARC) to switch from boosting cancer cell proliferation to inhibiting growth and inducing death [287]. Additionally, in a pancreatic cancer model, endothelial monocyte activating polypeptide II (EMAP II) inhibits fibronectin-integrin angiogenesis signalling, which results in a significant reduction in tumour development and a decrease in microvessel density and proliferative activity [288].

On the other hand, more drastic strategies have been proposed with the aim of eliminating or diminishing CAFs population. In this regard, FAPα inhibitors have shown promising preclinical results in reducing tumour progression and promoting an antitumour immune response [289]. In xenograft models of pancreatic, head and neck, and lung cancer in vivo, FAP5-DM1, a maytansinoid-conjugated monoclonal antibody, was demonstrated to suppress and produce complete regression of tumour growth [290]. Oral DNA vaccines against FAPα that have demonstrated efficacy in mice models of breast or colon carcinomas are also being tested [7], along with other immunotoxins targeting FAPα such FAP-PE38 [291] (Figure 6). Furthermore, it has been demonstrated that increasing CD8+ T cell antitumour responses using a chimeric antigen receptor (CAR) T-cell specific for FAPα can suppress the growth of a variety of subcutaneously implanted tumours in mice [292]. However, in mouse models of transgenic PDAC (KPC mice) and transplantable colon carcinoma (C26 cells), depletion of FAPα+ cells using the Diphtheria Toxin Receptor (DTR) system caused severe systemic toxicity, including cachexia and anaemia. This likely reflects the significance of FAPα+ stromal cells in maintaining normal muscle mass and haematopoiesis [199]. Given that fibroblasts play a significant role in numerous essential physiologic processes in humans, it is crucial to stress that completely eliminating entire fibroblast populations is quite difficult in patients. Furthermore, biomarkers like αSMA or FAPα are not only expressed by CAFs, which makes it more difficult to use them realistically.

GPR77 is an additional cell surface marker that has been researched in pre-clinical models. It has been demonstrated that this protein, along with CD10, can be used to pinpoint the fibroblast subpopulation that results in chemoresistance. It has been demonstrated that blocking GPR77 with a neutralizing antibody lowers the number of CSCs in tumours and improves the response to chemotherapy [267] (Figure 6). Furthermore, since these fibroblasts were discovered in the tumour before treatment, their simultaneous targeting with chemotherapeutic regimens may be advantageous for the patient.

These findings suggest that rather than eliminating CAFs entirely, effective anti-CAFs medicines for cancer would likely need to re-educate them towards a NAF phenotype or decrease CAFs number. Another factor to consider is the heterogeneity of CAFs subpopulations and, hence, current studies should aim at differentiating CAFs subtypes in order to therapeutically affect pro-tumoural CAFs and support antitumour CAFs. Notably, Olive et al. investigated whether co-delivery of the synthetic experimental medication IPI-926 (Figure 6), which suppresses stromal development by blocking Smo, a crucial step in the Hh pathway, could enhance the efficacy of GEM drug administration [293]. Reduced αSMA+ fibroblasts and pancreatic stromal depletion in the treated tumours improved GEM administration and lengthened life [293]. However, in a colon cancer model, activating Hh signalling in fibroblasts significantly decreased tumour burden and disease progression [294]. As a result, Hh-signalling CAFs appear to serve as both tumour promoters and suppressors, and it is still uncertain whether particular determinants predict this behaviour in various cancer types [295].

Furthermore, an enzymatic degradation of the ECM to decompress the tumour and to normalize vasculature has been proposed to promote a more efficient chemotherapeutic delivery to cancer cells in solid tumours. In this respect, hyaluronic acid (HA) degradation or anti-angiogenic therapies are under consideration. Examples include, among others, the angiotensin inhibitor losartan [296] and enzymatic therapies such as PEGPH20, a pegylated hyaluronidase enzyme designed to degrade HA produced by CAFs in TME. However, although freeing the stroma from its restrictive properties could improve the delivery of therapeutic compounds, PEGPH20 administration could also alter antitumour responses in TME, and while hyaluronidase treatment has shown promising results in animal models and early clinical trials [297,298], PEGHP20 in combination with nab-paclitaxel/GEM did not meet the criterion in phase III clinical trials in pancreatic cancer patients (Hakim et al., 2019).

MMPs derived from CAFs have also demonstrated their importance in inducing tumour growth, metastasis, and angiogenesis in animal model experiments [299]. Therefore, MMPs inhibitors could be another approach to therapeutically target MMPs and reduce enzymatic activity, but the reality is that MMPs inhibitors also do not meet phase III criterion at present [300,301].

CAFs reprogramming, as well as short-term inhibition of CAFs-predominant signalling pathways, have been recognized as better strategies than complete ablation or chronic inhibition. Indeed, several studies have found that strategies aimed at CAF ablation are ineffective [275,302].

Finally, the diverse therapeutic approaches emerging from basic and applied research are promising and encouraging. Unfortunately, because of the variability in the role of CAFs in each tumour, as well as their specific tissue source, they must be targeted at each tumour histotype. As a result, any future therapy targeting CAFs will be more effective after extensive research on their taxonomy and a precise and detailed understanding of the stroma–tumour relationship.

## 9. Conclusions and Future Prospects

Quiescent fibroblasts expand upon activation in response to stimuli released by damaged organs and emerging inflammation, generating growth factors and ECMs to autoregulate their expansion as well as to regulate inflammation and immunity. For decades, activated fibroblasts were thought to be cancer cells’ accomplices in promoting tumour growth. However, fibroblasts appear to act as involuntary regulators of tumour growth, either positively or negatively, depending on the origin of the fibroblast on one hand, and on the other hand, the context in which it is found, such as the stage of tumour progression [303].

To make things worse, CAFs exhibit remarkable heterogeneity and plasticity, allowing CAFs from the same tumour to arise through different pathways, have different phenotypes, and have functionally different effects on tumour cells and the microenvironment. Nonetheless, it is clear that controlling, reducing, or re-educating the CAFs subpopulation within the tumour is critical to improving current treatments. CAFs are appealing drug targets for a variety of reasons, including: (i) they are genetically more stable than neoplastic cells, making them less prone to developing resistant phenotypes due to high mutation rates and clonal selection; (ii) they have epigenetic differences from normal stromal cells; (iii) they are responsible for the structure of the tumour ECM; and, finally, (iv) CAFs support neoplastic cells throughout the disease spectrum, from pre-invasive lesions to the metastatic focus. Thus, by targeting CAFs, different stages of the disease, as well as numerous pathophysiological processes such as angiogenesis, EMT, and immune evasion, could be influenced.

All in all, the true role of tumour stroma in the invasion and metastasis of cancerous lesions remains unclear. In fact, experimental findings have supported both a protective and a permissive role against tumour development depending on the stage of tumour progression. Therefore, it is more than possible that fibroblasts are both “heroes” and “villains”, and it is necessary to investigate, in depth, the different factors involved in these interactions during the different tumour stages and to determine what causes fibroblasts to switch from one team to another and at what time.

## Figures and Tables

**Figure 1 ijms-23-15576-f001:**
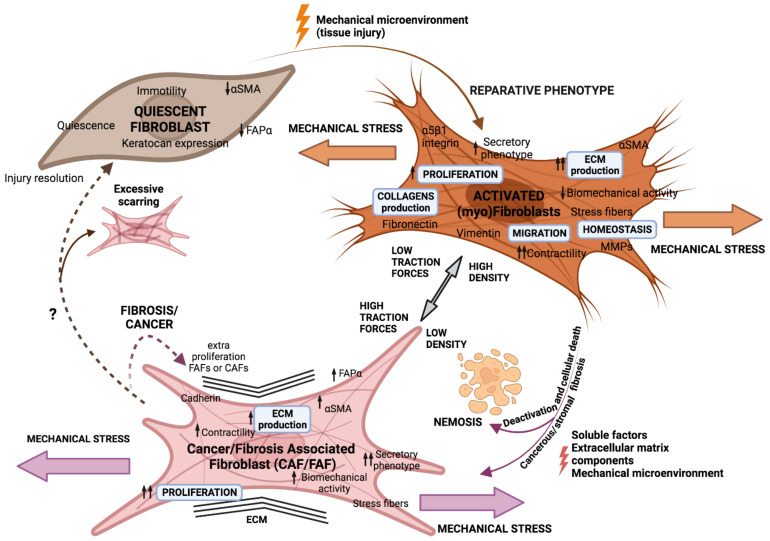
Fibroblasts’ phenotype adapts in response to external aggression. Activation of quiescent fibroblasts to a myofibroblast phenotype to repair tissue damage by wound healing is triggered by microenvironmental signals. Myofibroblasts should disappear by apoptosis (nemosis) after punctual aggression, but if fibroblasts are subjected to constant stress, they could evolve into a FAF or CAF phenotype.

**Figure 2 ijms-23-15576-f002:**
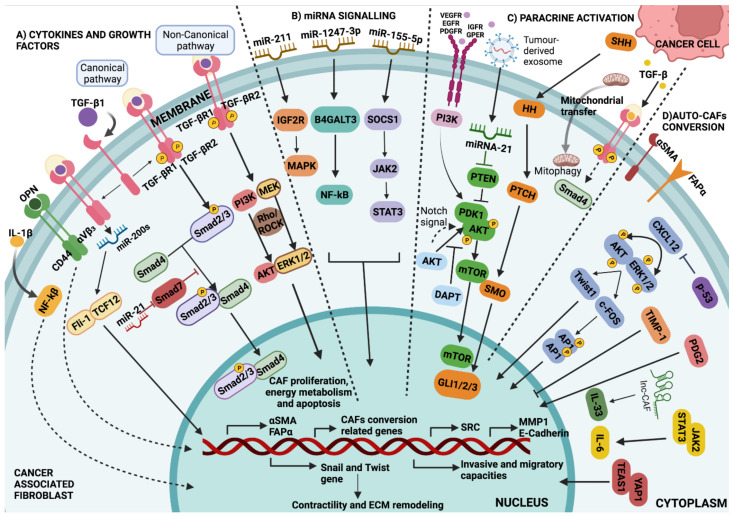
Major pathways involved in CAF development. (**A**) Cytokines and growth factors: Regulation of CAF-specific gene expression through TGF-β signalling via Smad-dependent and Smad-independent pathways. In the canonical pathway, Smad2/3-Smad4 translocate to the nucleus and bind to a specific DNA sequence, being possibly inhibited by Smad7. In the non-canonical pathway, TGF-β promotes the activation of several signalling pathways, other than Smad, including PI3K kinases, MEK, and Rho-Rock, among others. (**B**) miRNAs signalling: miRNAs and lncRNAs transform NAFs into CAFs through downstream signalling involving JAK/STAT, NF-κβ, and MAPK cascades. (**C**) Paracrine activation: Through the PDK1/AKT signalling pathway, in which tumour exosome miRNA-21 inhibits it and PI3K promotes it. Notch signalling via AKT signalling pathway controls AKT phosphorylation and mTOR activation. As a result, mTOR regulates the expression of targeted genes associated with differentiation into CAFs; in addition, expression of tumour cell-derived SHH has been confirmed to modulate CAFs via paracrine activation of HH signalling, and the overexpression of SMO in CAFs contributes to HH signal transduction and GLI1 activation. Fibroblasts can also be metabolically reprogrammed via cancer cell-derived mitochondrial transfer and the TGF-β signalling pathway derived from cancer cells. (**D**) Auto CAFs-conversion: A self-propelled conversion induced by changes in cellular homeostasis, which controls the activation of cytoskeletal proteins and the secretory phenotype via the YAP1/TEAS1 and JAK2/STAT3 signalling pathways. EMT is facilitated by an activated CXCL12 signal via the ERK/AKT-Twist1-MMP1 pathway. TIMP-1 is related to collagen contractility and CAF proliferation and migration through the ERK1/2 signalling pathway in CAFs (Table 1).

**Figure 3 ijms-23-15576-f003:**
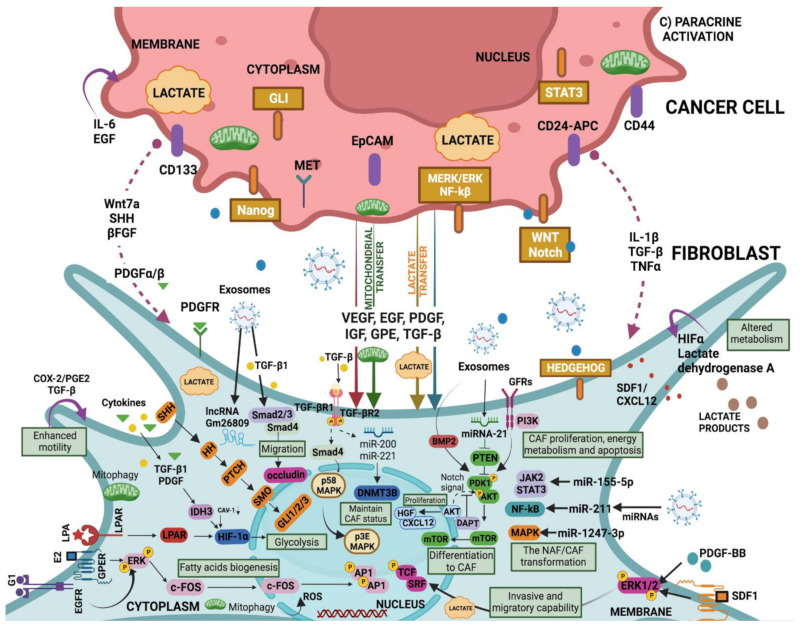
CAF “education” by the cancer-derived factors (extension of section C) in Figure 2). Cancer-derived exosomes carry elements such as miRNA and lncRNA that promote the transformation of NAF to CAF through downstream signals that include cascades such as TGF-β/Smads, JAK/STAT, NF-κB, and MAPK. NAF-CAF conversion may also be driven by the reprogramming of glucose metabolism and the HIF-1α signalling pathway involved in glycolysis. Both the canonical TGF-β signalling pathway (TGF-β/Smads) and the non-canonical pathway (with activation of TGF-β but not Smads) are actively involved in the malignancy of NAFs. In CAFs, while miRNA-21 can attenuate the inhibition of PTEN on PDK1/AKT, the receptor–ligand binding-activated PI3K can promote it. As a result, through the PDK1/AKT signalling cascade mTOR protein is transported into the nuclei, and subsequently, the mTOR protein regulates the expression of targeted genes associated with CAFs differentiation. Notch signalling pathway is also involved in CAF differentiation via AKT. In turn, CAF-mediated PI3K/AKT signalling pathway regulates cell proliferation, migration, and stemness in cancer cells. EGFR/ERK signalling in CAFs is stimulated by E2 and G1 and upregulates fatty acids metabolism. Further, PDGF-BB and SDF-1 stimulate a higher invasive and migratory capability of CAFs via ERK1/2 phosphorylation (Table 1).

**Figure 4 ijms-23-15576-f004:**
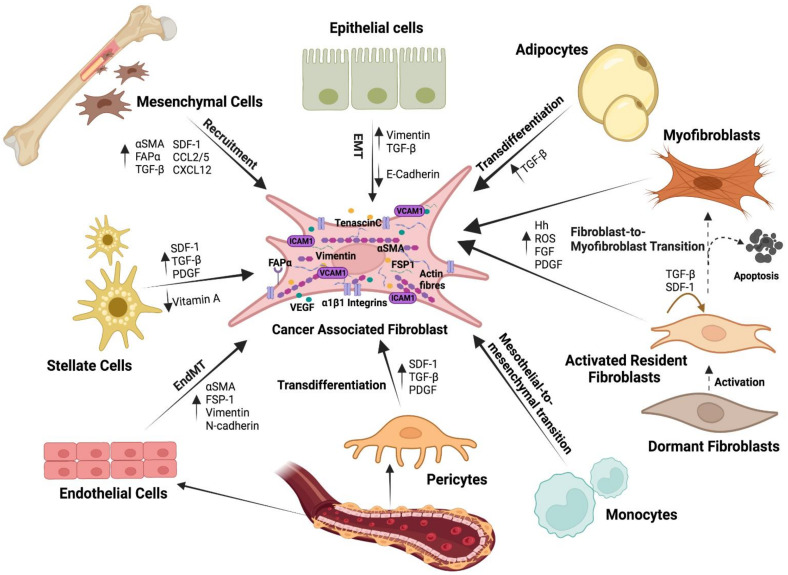
Heterogeneous cellular origin of CAFs. Within the TME, CAFs can be derived from a wide variety of cells: activated resident fibroblasts, bone marrow-derived mesenchymal stem cells, endothelial cells, epithelial cells, adipocytes, pericytes, and stellate cells. Depending on the origin, CAFs have distinctive characteristics, functions, and locations.

**Figure 5 ijms-23-15576-f005:**
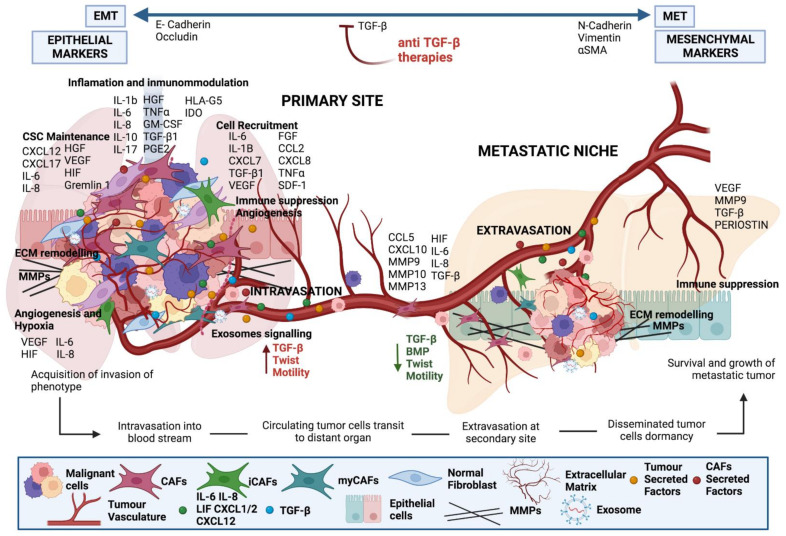
Crosstalk between tumour cells and CAFs at the primary tumour and metastatic site. Factors derived from cancer cells activate local fibroblasts to become CAFs. CAFs, in turn, produce interleukins to activate signalling in cancer cells increasing their metastatic potential. CAFs are chaperons of tumour cells as they support their survival and extravasation to the metastatic site, as well as preparing the premetastatic niche.

**Figure 6 ijms-23-15576-f006:**
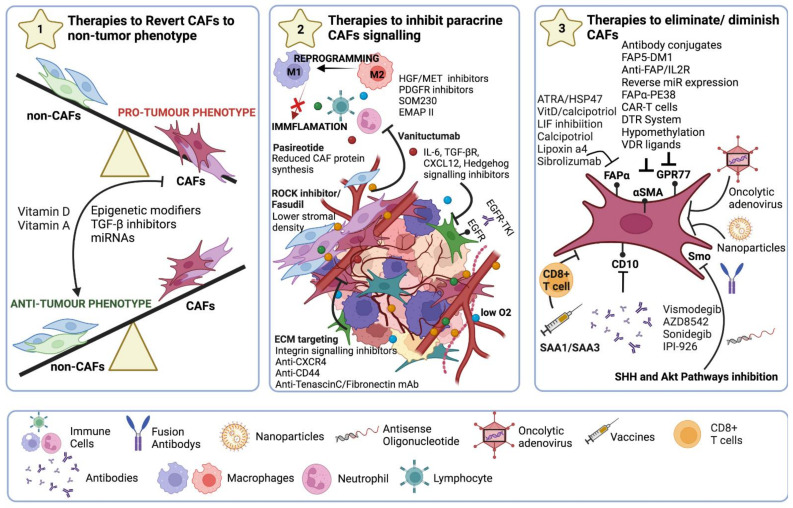
Strategies for CAF-targeted anticancer therapy. (**1**) Reprogramming to a non-CAF phenotype using epigenetic modifiers or TGF-β inhibitors. In addition, the use of anti-miRNAs, miRNA mimetics, or siRNA administration to reverse CAFs phenotype. (**2**) Blocking all signalling emitted by CAFs either by targeting crucial molecules (IL-6, TGF-β, CXCL12) or signalling pathways such as CCL2 and CCR2 signalling axis, JAK-STAT3, TGF-β, and the Hedgehog signalling pathway, or FAK signalling pathways to restrict ECM remodelling along with TN-C, MMPs, and HA. (**3**) Targeting αSMA or membrane markers such as FAPα, GPR77, or CD10 to eradicate CAFs populations.

**Table 1 ijms-23-15576-t001:** Tumour cell’s secreted factors and their functions over CAFs.

Factors	Mode of Action	References
Cytokines	CAF activation/inactivation	[4,22,23,24,32,33,34,35,36,37,38,39,40,41,42,43,44,45]
IL-1	CAF activation: Promotes cell growth and invasion by NF-κB and IL-1β/IL-1R pathway	[8,45,46,47]
TGF-beta	CAF activation: Promotes cell proliferation and migration by TGF-β/Smad pathway	[22,37,38,39,40,41,42,48,49,50,51,52,53,54,55,56,57,58,59,60,61]
IL-6	CAF activation: Promotes cell chemoresistance, cancer progression, cell metastasis and chemoresistance and cell adhesion by STAT3/NF-kbeta, JAK2/STAT3, ERK1/2, STAT3/Notch, and STAT3/PD-L1 pathways	[23,24,36,39,46,50,59,62,63,64]
PDGF	CAF activation	[39,40,48,49,65,66,67,68,69,70,71,72,73,74,75,76,77,78]
EGF	CAF activation	[49,79,80,81]
IGF	CAF activation: Promotes cell chemoresistance, cell survival, and cell growth by IGF-/AKT, IGF/IGFR, and PI3K/AKT/mTOR pathways	[49,82,83,84,85]
JAK2/STAT3	CAF activation	[40,86]
YAP1/TEAS1	CAF activation	[87,88]
p53	CAF inactivation	[89]
VEGF	CAF activation	[51,75,76,90,91]
CXCL12	CAF activation: Promotes cell metastasis, cell invasion, EMT and cisplatin resistance and cancer progression by PI3K/AKT, TGF-β, Wnt/β-catenin, and CXCL12/CXCR4 pathways	[32,33,34,35,70,83,92,93,94,95,96,97,98,99]
TIMP-1	CAF inactivation	[100,101,102,103]
OPN	CAF activation	[44,53,79,104,105,106,107,108,109,110]
MAPK	CAF activation	[44]
AKT	CAF activation	[44,98]
ERK1/2	CAF activation	[109,110]
c-Ski	CAF inactivation	[111]
IL-33	CAF activation: Promotes cell migration and invasion	[112]
TNF	CAF activation	[113]
SHH	CAF activation	[114]
miRNAs	CAF activation/inactivation	[115,116,117,118,119]
miR-200s	CAF activation: Invasion, metastasis via transcription factors Fli-1 and TCF12	[70]
miR-21	CAF activation: miR-21 and Smad7 induce CAF by TGF-β1 signalling regulation. Motility and invasion by MMP inhibitor RECK	[120,121]
miR-155-5p	CAF activation: Angiogenesis by SOCS1/JAK2/STAT3 signalling pathway	[122]
miR-1247-3p	CAF activation: Stemness, EMT, chemoresistance, and tumorigenicity by IL-6/8; lung metastasis by β1-integrin-NF-κB pathway	[123]
miR-211	CAF activation	[124]
lncRNA	CAF activation/inactivation	[125,126,127]
LPA	CAF activation	[128]
GPE	CAF activation: Promotes cell proliferation by GPER/EGFR/ERK pathway	[129]
DAPT	CAF inactivation	[130]
Tumour-derived exosome	CAF activation	[131]
Mitocondrial transfer	CAF activation	[132]
SMAD	CAF activation/inactivation	[121,127,133,134]
Smad2/3	CAF activation	[134,135]
Smad4	CAF activation	[127,133,134]
Smad7	CAF inactivation	[121]

Green stands for CAFs’ activators factors; red for CAFs’ inactivators factors; and yellow for factors with dual function (activators and inactivators).

## Data Availability

Not applicable.

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
