# Peer review of "Dual Role of Fibroblasts Educated by Tumour in Cancer Behavior and Therapeutic Perspectives"

_ijms, 2022, doi:10.3390/ijms232415576_

Round 1

Reviewer 1 Report

The authors have provided an extensive overview of the different types of fibroblasts and their multiple potential origins. They have discussed their role both as protective (heros) and as tumor-progressive (villains) agents. The authors have appropriately discussed the complex cross-talk happening between tumor parenchyma and tumor stroma and the integral role fibroblasts play in orchestrating it.

Overall, the review is quite nicely written covering multiple facets like recruitment of fibroblasts at neoplastic sites, influence of different growth factors/cytokines on their conversion into cancer associated fibroblasts (CAFs), different surface markers being expressed at that stage, their contribution towards tumor immunity  and therapeutic resistance. The authors end the review by citing different therapeutic strategies that have been used to either re-educate, reprogram or eliminate CAFs.

This is a relevant and timely review that provides comprehensive insights to its readers about the complex role of stroma and its education at the tumor site.

Few minor comments are:

- Line 35/36: Can the authors give actual number of cancer deaths in 2020 rather than an expected number? Or estimate it for the year 2022!

- Line 240: Reconsider sentence construction

- Line 544: T-cell anergy (not energy!)

- Line 593: detection of novel (not novels!)

Author Response

Point-by-Point Response to Reviewers

Response to Reviewer 1 Comments

The authors have provided an extensive overview of the different types of fibroblasts and their multiple potential origins. They have discussed their role both as protective (heros) and as tumor-progressive (villains) agents. The authors have appropriately discussed the complex cross-talk happening between tumor parenchyma and tumor stroma and the integral role fibroblasts play in orchestrating it.

Overall, the review is quite nicely written covering multiple facets like recruitment of fibroblasts at neoplastic sites, influence of different growth factors/cytokines on their conversion into cancer associated fibroblasts (CAFs), different surface markers being expressed at that stage, their contribution towards tumor immunity  and therapeutic resistance. The authors end the review by citing different therapeutic strategies that have been used to either re-educate, reprogram or eliminate CAFs.

This is a relevant and timely review that provides comprehensive insights to its readers about the complex role of stroma and its education at the tumor site.

We appreciate the encouraging comments of Reviewer 1 on our manuscript and we have notably improved the quality of this manuscript according to his/her suggestions. 

Point 1: Line 35/36: Can the authors give actual number of cancer deaths in 2020 rather than an expected number? Or estimate it for the year 2022!

Response 1: Following Reviewer's suggestion, the following text has been added to the manuscript: “According to the American Cancer Society, in 2022 there will be an estimated number of 1.9 million new cancer cases diagnosed and 609,360 cancer deaths in the United States, according to these data, cancer is, and will continue to be, one of the worst diseases in the world.” (PAGE: 2, PARAGRAPH: 2, LINE: 35/36)

Point 2: Line 240: Reconsider sentence construction

Response 2: In agreement with Reviewer 1 the sentence has been rewritten:
“In the first place, we identify αSMA, a cytoskeleton protein associated with TGF-β production, with a highly contractile phenotype [78] and which is considered a robust marker of CAFs possessing myofibroblast morphology [79]”. (PAGE: 8, PARAGRAPH: 2).

Point 3: T-cell anergy (not energy!)

Response 3: We appreciate Reviewer 1’s relevant comment on our manuscript and have corrected the incorrect expression. (PAGE: 14, PARAGRAPH: 1).

Point 4: detection of novel (not novels!)

Response 4: We thank Reviewer 1 for having corrected this grammatical mistake. (PAGE: 14, PARAGRAPH: 3).

Reviewer 2 Report

In this review, the authors focused on the interaction between CAF, tumor, and tumor microenvironments in detail. Moreover, many essential and latest findings are summarized as good figures. This manuscript is significant and informative; however, I raised several points to improve the report's content. 

Query

1. In the figures, the authors explain the external stimulation for CAF activation or inactivation. Of course, the stimulation from cancer to CAF has already been mentioned. However, although the authors focus on cancer-induced fibroblast education in their title, their description is limited. A table summarizing the various fibroblast stimulators already introduced in the figures derived from cancer would do the review more in line with the title. One figure should be used to explain the CAF education by the cancer-derived factors because the explanation in the upper right corner of Figure 2 was insufficient.

2. Some keywords in the figures were not described and explained in the manuscript. For instance, who reported the function of miR-12477-3p in CAF development? I could not find the description regarding the microRNA in the main document. I think that the review's figures are excellent; however, it is not easy to check whether the citation is suitable or not by the present documents because some crucial keywords are not used in the main documents. This point should be improved for readers who may cite this review in the future.

3. The information of all keywords and reference papers should be linked as table data. If the factors are cancer-derived activators or inactivators of CAF, this vital information should be explained in the table.

4. Has this paper been proofread in English? If there are suspected errors in the figures, such as the miR-12477-3p described earlier, the standard English proofreading process may not check highly specialized terms. Therefore, authors should repeatedly check all technical terms (especially in the figures).

Author Response

Point-by-Point Response to Reviewers

Response to Reviewer 2 Comments

In this review, the authors focused on the interaction between CAF, tumor, and tumor microenvironments in detail. Moreover, many essential and latest findings are summarized as good figures. This manuscript is significant and informative; however, I raised several points to improve the report's content.

We welcome Reviewer 2’s positive comments on our manuscript and have changed the text according to his/her suggestions in order to improve the report’s content.

Point 1: In the figures, the authors explain the external stimulation for CAF activation or inactivation. Of course, the stimulation from cancer to CAF has already been mentioned. However, although the authors focus on cancer-induced fibroblast education in their title, their description is limited. A table summarizing the various fibroblast stimulators already introduced in the figures derived from cancer would do the review more in line with the title. One figure should be used to explain the CAF education by the cancer-derived factors because the explanation in the upper right corner of Figure 2 was insufficient.

Response 1: According to Reviewer 2’s suggestion we have added a table summarizing cancer derived factors that are included in the figures (Table 1).

In addition, following Reviewer’s suggestion we have added a new figure summarizing molecular pathways related to CAF induction by the cancer-derived factors (Figure 2.2).

Point 2: Some keywords in the figures were not described and explained in the manuscript. For instance, who reported the function of miR-12477-3p in CAF development? I could not find the description regarding the microRNA in the main document. I think that the review's figures are excellent; however, it is not easy to check whether the citation is suitable or not by the present documents because some crucial keywords are not used in the main documents. This point should be improved for readers who may cite this review in the future.

Response 2: We thank Reviewer 2 for having raised this important point and it is that due to the great magnitude of information discussed in the figures, for a better understanding, it has been included in the main text only the main factors included in the Figures.

Following Reviewer 2’s suggestion, references have been included in Table 1, with the factors cited in the figures and their appropriate references, in addition, references have been reviewed though the text.

Point 3: The information of all keywords and reference papers should be linked as table data. If the factors are cancer-derived activators or inactivators of CAF, this vital information should be explained in the table.

Response 3: In agreement with Reviewer 2’s a Table has been added (Table 1) in the manuscript, including key cancer-derived factors which are CAF activators (green), inactivators (red) or both (yellow).

Point 4:  Has this paper been proofread in English? If there are suspected errors in the figures, such as the miR-12477-3p described earlier, the standard English proofreading process may not check highly specialized terms. Therefore, authors should repeatedly check all technical terms (especially in the figures).

Response 4: We thank and agree with Reviewer 2, and following Reviewer’s guides we have corrected miR-12477-3p in Figure 2 (miR-1247-3p) and the whole text have been proofread in order to avoid mistakes in highly specialized terms.

Round 2

Reviewer 2 Report

This manuscript is a re-submission for publication. The authors provide additional information and do a good job of answering many of the reviewer's critiques of their initial submission. This report is improved and will meet the criteria of this journal.